# Molecular Docking Guided Grid-Independent Descriptor Analysis to Probe the Impact of Water Molecules on Conformational Changes of hERG Inhibitors in Drug Trapping Phenomenon

**DOI:** 10.3390/ijms20143385

**Published:** 2019-07-10

**Authors:** Saba Munawar, Jamie I. Vandenberg, Ishrat Jabeen

**Affiliations:** 1Research Center for Modeling and Simulation (RCMS), National University of Science and Technology, Sector H-12, Islamabad 44000, Pakistan; 2Victor Chang Cardiac Research Institute, Lowy Packer Building, 405 Liverpool Street, Darlinghurst Sydney, NSW 2010, Australia

**Keywords:** hERG, GRIND, acquired LQTs, 3D QSAR, molecular docking simulations

## Abstract

Human ether a-go-go related gene (hERG) or KV11.1 potassium channels mediate the rapid delayed rectifier current (*I_Kr_)* in cardiac myocytes. Drug-induced inhibition of hERG channels has been implicated in the development of acquired long QT syndrome type (aLQTS) and fatal arrhythmias. Several marketed drugs have been withdrawn for this reason. Therefore, there is considerable interest in developing better tests for predicting drugs which can block the hERG channel. The drug-binding pocket in hERG channels, which lies below the selectivity filter, normally contains K^+^ ions and water molecules. In this study, we test the hypothesis that these water molecules impact drug binding to hERG. We developed 3D QSAR models based on alignment independent descriptors (GRIND) using docked ligands in open and closed conformations of hERG in the presence (solvated) and absence (non-solvated) of water molecules. The ligand–protein interaction fingerprints (PLIF) scheme was used to summarize and compare the interactions. All models delineated similar 3D hERG binding features, however, small deviations of about ~0.4 Å were observed between important hotspots of molecular interaction fields (MIFs) between solvated and non-solvated hERG models. These small changes in conformations do not affect the performance and predictive power of the model to any significant extent. The model that exhibits the best statistical values was attained with a cryo_EM structure of the hERG channel in open state without water. This model also showed the best R^2^ of 0.58 and 0.51 for the internal and external validation test sets respectively. Our results suggest that the inclusion of water molecules during the docking process has little effect on conformations and this conformational change does not impact the predictive ability of the 3D QSAR models.

## 1. Introduction

Over the last two decades, a vast array of structurally and functionally unrelated drugs including antiarrhythmics (dofetilide) [1], antibiotics (grepafloxacin) [2,3], antipsychotics (sertindole, haloperidol) [3] and antihistamines (astemizole) [4] have been withdrawn from the market due to their association with prolongation of the ventricular action potential that causes acquired long QT syndrome (aLQTs) [5]. The aLQTs is characterized by QT interval prolongation on the surface of an electrocardiogram (ECG) that may lead to fatal ventricular arrhythmias, that is, Torsade de Pointes (TdPs) [6]. The major cause of drug-induced QT interval prolongation is the trapping of drugs in the pore of the human ether-a-go-go-related gene (hERG) or KV11.1 potassium ion channel [7]. hERG encodes for the α-subunit of the rapid component of delayed rectifier current (*IKr*), a voltage-gated K^+^ ion channel that plays a pivotal role in the repolarization phase of the cardiac action potential [7]. It has been estimated that 60% of the drugs in the development phase show hERG liability [8], 15% of drugs on the market was associated with a tendency to prolong QT interval and 4% was associated with TdPs (www.crediblemeds.org). Subsequently, to assess the pro-arrhythmic risk of new chemical entities (NCEs), the regularity authorities stipulate pre-clinical safety guidelines that all drugs must be tested for their hERG liability and tendency to induce QT interval prolongation [9,10]. Additionally, the International Conference on Harmonisation (ICH) has recommended a thorough QT/QT_calculated_ study for any bioavailable drug before marketing [11]. 

The hERG channel can adopt open, inactivated or closed conformational states, like other voltage-gated channels. However, in comparison with other K^+^ channels hERG exhibit highly promiscuous binding cavity because of structural peculiarities [12]. Previously, molecular docking and molecular dynamics (MD) studies were performed using homology modeling of different conformational states of hERG [13,14,15,16,17,18]. Various in silico tools and applications based on machine-learning methods and QSAR models are available online. These models are trained with different molecular descriptors. These applications include StarDrop (http://www.optibrium.com/stardrop/), QuikProp from Schrodinger Suite (Schrodinger, LLC, http://www.schrodinger.com/), AdmetSAR (http://lmmd.ecust.edu.cn:8000), and the Pred-hERG web app (http://www.labmol.com.br/predherg). Due to unavailability of the highly resolved X-ray structure of hERG along with a prototype ligand and its promiscuous binding cavity, predicting correct binding conformations of ligands and building effective and reliable in silico models remains a big challenge [19]. Wang and MacKinnon recently determined the cryo-EM structure of hERG with the resolution of 3.8 Å in an open conformational state [20]. Although the resolution of the structure is not sufficient to permit elucidation of drug binding poses, it is the only experimental structure available to date. The open conformational state structure revealed the sprouting of hydrophobic pouches from the central cavity, which may play a critical role in the promiscuity of hERG drug binding. Additionally, the presence of water molecules within the binding cavity of the hERG channel has not received sufficient attention in investigations of drug binding to these channels, yet water molecules could have important implications for which drug binding poses and side-chain orientations are most favorable in terms of binding free energy [21,22,23].

Previously, it was reported that during in silico drug design process, water often complicates the calculations for finding binding affinities and final poses. Including water, may, however, help to improve binding by balancing the enthalpy and entropy change, obtaining favorable binding energies and probable binding poses [21,22] and improve molecular dynamics-based binding affinities [24,25]. Reports on the current docking protocols have various issues including protein flexibility, treatment with water molecules and evaluation of binding free energies that are not yet solved satisfactorily [26]. The presence of water molecules affects the binding modes and subsequently, these conformations affect the prediction of in silico QSAR models [19,27].

The binding cavity of hERG can accommodate up to 50 water molecules, however, the role of water towards the drug-trapping phenomenon of hERG is currently unknown. After the availability of the first cryo structure of hERG, there is considerable interest, especially in addressing the question of whether water molecules have any impact on the drug-trapping phenomenon in general and molecular conformations of drugs in particular. In this paper, we developed various Grid-Independent Descriptor (GRIND) models using conformations obtained by solvated and non-solvated hERG channel in its open and closed conformational state. GRIND are alignment independent but conformational dependent descriptors therefore, different conformations can affect the model performance and predictive ability [28] of the model. Here, we elucidate the impact of water molecules on the conformational change of hERG blockers as compared to the non-solvated channel.

## 2. Results

### 2.1. Molecular Docking Simulations

A curated database of 207 structurally diverse hERG inhibitors as described by Munawar et al. [29] was divided into 166 training (80%) and 41 test compounds (20%) by the diverse subset selection method (see Appendix A). Two test sets namely internal test set (20% compounds of the whole data set) and external test set (latest hERG inhibition data gathered from literature. Please see Section 4 for detail) were used for validation of the models.

The training set was used to dock in an open state of hERG cryo_EM structure (PDB ID: 5va1) [20] and close conformational state of hERG homology model based on kcsA template [15]. In order to obtain a stable cryo_EM structure, a MD simulation run of 1 ns with 50,000 iteration steps of minimization was performed using CHARRM22 force field [30] in Not Another Molecular Dynamic (NAMD) software [31] as discussed in the Section 4. The minimization plot shows that a stable structure is achieved after 500 steps; however, to further validate minimization was performed up to 50,000 steps (Appendix A). The root mean square deviation (RMSD) of the Cα atom from the initial coordinates with respect to time. The Cα RMSD plot shows that a stable hERG cryo_EM structure was achieved at 0.6 Å (Appendix A). To probe the impact of water molecules on hERG binding, both conformational states of hERG were solvated by setting up a water box of X: 75.019 Y: 74.702 and Z: 90.738 around centroid point of binding cavity. Thus, four independent sets of molecular docking solutions were generated. These include molecular solutions (poses) in (1) non-solvated hERG cryo_EM structure (Figure 1A,B), (2) solvated hERG cryo-EM structure (Figure 1C,D), (3) non-solvated hERG homology model in closed state (Figure 2A,B) and (4) solvated hERG homology model in closed state (Figure 2C,D) designated as non-solvated-open, solvated-open, non-solvated-closed, and solvated-closed respectively. 

The binding free energy of docked ligands within non-solvated-open, solvated-open non-solvated-closed and solvated-closed hERG binding cavities was estimated using GBVI/WSA dG scoring function. It estimates the sum of the van der Waals, electrostatic, and solvation energies, using generalized born solvation model (GB/VI) [32]. The binding free energy scores of resultant docking solutions vary from −226 to −621 kcal/mol for molecular non-solvated-open, −146 to −476 kcal/mol for solvated-open, −420 to −1365 kcal/mol for non-solvated-closed and −256 to −622 kcal/mol for solvated-closed. Plots in Figure 3 show the binding free energy of the compounds of entire training dataset in non-solvated-open, solvated -open (Figure 2A) and non-solvated-closed and solvated-closed (Figure 2B) models of hERG. 

Briefly, the binding free energy scores of ligand poses within solvated cavities are comparatively higher than non-solvated poses (Figure 3). The higher energy values of solvated complexes might be due to the entropic contribution of desolvation effect as it was reported previously that water desolvation does impact on enthalpy and entropy of the binding complex [33]. Overall, the most probable binding pose of each ligand was selected on the basis of the least energy score and used for further ligand–protein interaction analysis. The ligand–protein interaction profiles of prototype hERG inhibitors including MK-499, E4031, dofetilide and trimethoprim as shown in Figure 4 were selected to get deeper insights into the binding in the solvated and non-solvated hERG cavity. 

The ligand–protein interactions of archetypical hERG blockers in non-solvated-open and solvated-open state of hERG are shown in Figure 5. MK-499 in non-solvated hERG-binding cavity showed a hydrogen bond acceptor interaction with Gly648 whereas a hydrogen bond donor interaction was observed with Ser624 and Ser660. Moreover, hydrophobic or π–π interactions were observed between the aromatic rings of the MK-499 and Tyr652 as shown in Figure 5A. Similarly, in solvated binding cavity MK-499 showed hydrogen bond acceptor interaction with Leu622. The oxygen of methanesulfonamide group formed ligand–water (L–W) hydrogen bonding with water molecules (Figure 5B). However, no water-mediated interaction with any amino acid residue of the binding cavity was observed. E4031 in non-solvated-open binding cavity showed hydrogen bond donor interaction with Leu622 and π–π interactions with Tyr652 (Figure 5C). However, in solvated open binding cavity ligand–water–receptor (L–W–R) or water-mediated interaction was observed. The water molecule was bridging between the carboxylic group of E4031 and Ser624. The π–π interaction was also found between the benzene ring of E4031 and Ser624 as shown in Figure 5D. Pairwise analysis of energy scores of the solvated and non-solvated docked complex of E4031 revealed a lower energy score (favorable) value of −487 kcal/mol for non-solvated complex as compared to solvated complex (−387 kcal/mol) for which the energy score difference is 100 kcal/mol as shown in Table 1. The higher energy score of the solvated complex might delineate the entropic contribution due to desolvation effect during ligand binding within the solvated hERG cavity. Dofetilide showed hydrogen bond acceptor interaction with Ser621, Ser649 and Thr623 in non-solvated cavity (Figure 5E) whereas in solvated state methanesulfonamide group showed hydrogen bond interaction with Ser649 and Ser621 (Figure 5F). Moreover, Ser624 forms hydrogen bond donor interaction with the basic nitrogen through water-mediated L–W–R interactions. Similar to E4031, the slightly higher energy score (unfavorable) of dofetilide (−385 kcal/mol) in solvated hERG cavity was observed as compared to non-solvated (−397 kcal/mol) as listed in Table 1. Some of the compounds in Figure 3 show energy values less than 1000 kcal/mol that might be due to the limitation of Generalized-Born Volume Integral (GBVI) model that accounts for the presence of water–solvent hydrogen bonds in the implicit solvent model only approximately, at a mean-field level, which may under- or overestimate their strengths in some specific cases. Therefore, the solvation free energy approximation qualitatively reproduces the energy values, rather than quantitatively. 

Trimethoprim showed hydrogen bond interaction with Ser624 and Thr623 in the non-solvated cavity (Figure 5G), however, in the solvated cavity, the π–π interaction was observed between the aromatic ring of trimethoprim and Tyr652. The hydrogen bond interaction of ligand–water (L–W), water–water (W–W), and water–receptor (W–R) were also observed between trimethoprim and Tyr652 as shown in Figure 5H. Almost similar residues showed interactions within solvated and non-solvated binding cavities of hERG however, in the presence of water docked conformations, trimethoprim showed higher energy scores and thus, reflect less favorable hERG conformations as compared to non-solvated complexes. Interestingly, none of the selected pose (on the basis of least energy values) of prototype ligands was making interaction with Phe565. Only 3% poses of training dataset were found to interact with one of the Phe656. In order to further analyze the overall interaction profile of all docked ligands with solvated and non-solvated-open cavities of hERG, protein–ligand interaction fingerprint (PLIF) method (as discussed in the Section 4) was used.

Protein ligand interaction fingerprints (PLIF) in Figure 6 represent a population histogram summarizing the frequency of occurrence of interactions between training dataset of 166 compounds and non-solvated open and solvated open binding states of hERG. The PLIF analysis of training data compounds in open conformational state of hERG revealed the presence of hydrogen bond acceptor (HBA), hydrogen bond donor (HBD), hydrophobic, π–π interactions, and surface contact with Thr623, Ser624, Tyr652, and Phe656 amino acid backbone and side chain amino acid residues. Overall, 35% of the docking conformations of training data in non-solvated-open hERG cavity showed hydrogen bond acceptor interaction with side chain residue Ser624. However, in solvated-open cavity Ser24, Val625, Gly648, and Tyr652 mostly form water mediated interaction with the drugs and therefore, the overall frequency of direct hydrogen bond interactions is reduced (Figure 6). However, 70% of the finally selected docking solutions of all docked ligands showed π–π interactions with Tyr652 in solvated-open hERG cavity which further highlights the desolvation of water before hydrophobic interaction of ligand within the hERG cavity.

Similarly, the ligand–protein interaction profiles of selected hERG inhibitors in non-solvated-closed and solvated-closed hERG model are shown in Figure 7. The final docking pose of MK-499 in non-solvated-closed cavity of hERG showed π–π interaction with Tyr652 and hydrogen bond acceptor interaction with Gly648 (Figure 7A). Whereas in the solvated-closed binding cavity the carbonitrile group of MK-499 forms ligand–water (L–W) hydrogen bond interaction. Moreover, a water-mediated interaction between methanesulfonamide group of MK-499 and Gly648 was also observed (Figure 7B). Overall, the selected binding pose of MK-499 in the solvated-closed hERG cavity represents energetically less favorable score (−467 kcal/mol) as compared to the binding score (−949 kcal/mol) within the non-solvated closed state (Table 1). E4031 in non-solvated closed cavity showed hydrogen bond acceptor interaction with Ser649 (Figure 7C) however, in solvated closed cavity hydrogen bond acceptor interaction with Ser649 and π–π interaction with Tyr652 was observed. The oxygen of the methanesulfonamide group was involved in making hydrogen bond with water molecules present in closed binding cavity. The ligand–water (L–W) and water–water (W–W) interactions were also observed (Figure 7D). The final docked pose of dofetilide in the non-solvated-closed state was involved in making hydrogen bond acceptor interaction with Thr623 and Leu622 (Figure 7E) whereas in solvated state hydrogen bond acceptor interaction with Thr623 and ligand–water interaction was observed (Figure 7F). Similarly, final pose of Trimethoprim showed hydrogen bond acceptor interaction with Leu622 and π–π interaction with aromatic rings of Tyr652 in non-solvated-closed state binding cavity of hERG (Figure 7G) however, in solvated-closed binding cavity hydrogen bond acceptor interaction with Leu622 and water-mediated interaction between nitrogen of ligand and water molecule (L–W) and water molecule with another water molecule (W–W) are observed as shown in Figure 7H. The analysis of the final selected ligands showed that solvent contact is more frequent in closed binding cavity rather than open binding cavity. This might be due to abundance of water molecules in closed cavity (Figure 2D) as compared to open state binding cavity (Figure 1D). Furthermore, PLIF (Figure 8) summarizes the overall interactions present between the training dataset and non-solvated and solvated closed hERG cavities.

The population histogram in Figure 8 summarizes the overall interaction profile of the finally selected docked poses of the training data in solvated and non-solvated-closed hERG cavities Overall, hydrogen bond interactions with Thr623, Ser649, Leu622, and Ser624 were observed (Figure 8). Additionally, frequency of π–π interactions with Tyr652 and Phe656 is significantly reduced and water-mediated interactions were more frequently observed in both solvated and non-solvated-closed hERG cavities as compared to the solvated and non-solvated-open cavities. Thus, the presence of excess of water molecules in closed state of hERG might not allow direct contact of compounds with the hydrophobic binding residues Tyr652 and Phe656.

Results of protein–ligand interaction profiling and binding energies showed that adding explicit water molecules does not improve the binding free energies and docking scores. For further verification, docked conformations using GoldScore fitness function were also generated. However, in order to further validate our results and to correlate conformational differences of ligands in solvated and non-solvated hERG cavity with the predictive ability of the 3D QSAR model, grid-independent molecular descriptor (GRIND) analysis was performed using already defined eight independent sets of docked conformations. GRIND is conformational dependent approach therefore, any conformational change consequently may impact on statistical parameters, quality, performance and predictive ability of the model [34].

### 2.2. Grid-Independent Descriptor Analysis 

The GRIND is a novel set of alignment independent descriptors that has the ability to develop predictive 3D QSAR model for structurally diverse and large datasets. The 3D structural coordinates of the data are converted into Euclidean distance. The GRIND models are more reliable and robust and have better predictive ability compared to other 3D QSAR models [28]. Eight independent 3D QSAR models were developed by docked conformations generated using non-solvated-open, solvated-open, non-solvated-closed, and solvated-closed cavity of hERG channel using GoldScore fitness function (Gold v 5.3) and GBVI/WSA dG scoring function (MOE I 2018). The GRIND variables correlate with each conformational set of the training data with biological activities (pIC_50_) using partial least square analysis (PLS) of leave one out (LOO) cross-validation procedure [35] in software Pantacle v1.07 [36]. Table 2 summarizes the predictive ability (q^2^)^,^ correlation coefficient (r^2^) and standard deviation of error of prediction (SDEP) of all eight models obtained with different molecular conformations. In order to remove inconsistent variables, the fractional factorial design (FFD) algorithm was applied to raw variable data [37]. The best model that exhibits statistically significant parameters of q^2^ = 0.54, r^2^ = 0.62, and SDEP of 0.98 (Table 2) was attained at latent variable 2 (LV2) with docked conformations generated using GBVI/WSA dG scoring function implemented in MOE (I 2018) in the non-solvated open state of hERG. Model statistics results (Table 2) revealed that in comparison to GoldScore the GBVI/WSA dG scoring function deals better with the solvation and binding free energy. Therefore, for internal and external test set validation analysis only those models were considered that were build using GBVI/WSA dG docked conformations. Figure 9 illustrates observed versus predicted (pIC_50_ values) plot of the training set for these models.

The residual difference between observed versus predicted pIC_50_ values was of a maximum of ±1.5 log unit as shown in internal and external test set predictions provided in Appendix A. The final model of each conformational set was saved for further comparison and external validation procedure. 

The partial least square (PLS) coefficient plots of GRIND variables were obtained at LV-2 (latent variable) for each model after FFD1, as shown in Figure 10. The height and consistency of the peaks showed the presence of respective variables and their positive or negative contribution towards hERG inhibition. The PLS-coefficient plot of all four models in Figure 10 shows a slight variability in heights and consistencies of the peaks that might depict the small molecular conformational differences in solvated and non-solvated models. 

Briefly, four different probes including, ‘DRY’ (hydrophobic), ‘N1′ as (neutral flat amide: hydrogen bond donor), O (sp2 carbonyl oxygen: hydrogen bond acceptor), and TIP (molecular shape) were used to compute molecular interaction fields (MIF). Overall, the DRY-DRY (yellow), DRY_TIP (green) and DRY-O (brown) variable peaks show positive impact towards hERG inhibition, however, N1-N1 (blue) and O-N1 (turquoise) peaks show negative contribution towards hERG inhibition. The O-O (red) peak that showed the distance between two hydrogen bond donor moieties is slightly different in the model developed by using ligands conformations in non-solvated-open hERG cavity as compared to other models. This peak depicts that in molecular conformations produced in non-solvated-open hERG cavity, two hydrogen bond donor groups when present at a shorter distance (2.0 to 2.4 Å), reduce the hERG inhibition potential of molecules (negative peak). However, when these two groups are present at a larger distance of (16.0–16.4 Å) they increase the hERG inhibition potential (positive peak). Except for this O-O feature in solvated cryo_EM model almost similar pattern of positive and negative contributors was observed in solvated and non-solvated models (Figure 10). Table 3 enlists the hot spot indication of each feature along with distances and the difference between different states.

Briefly, DRY-DRY (yellow peaks) is the more pronounced feature in all four models showing the mutual distance between two hydrophobic moieties. The MIF analysis revealed that a distance of 12.0–12.4 Å in the non-solvated-open and 12.4–12.8 Å in the solvated-open state was found between hotspots as shown in Figure 11A and Figure 11B. This showed that conformations deviate approximately ~0.4 Å between non-solvated open and solvated open conformations and explicit water molecules have only a slight impact on the conformations. Similarly, DRY-TIP (yellow–green) and DRY-O (yellow–red) showed positive impact on hERG inhibition in both non-solvated and solvated states. However, selected molecular conformations in the non-solvated and solvated model deviate ~0.4 Å (Figure 11 and Figure 12). The N1-N1 (blue) and O-N1 (red-blue) features at a particular distance (Table 3) impact negatively towards hERG inhibition, however, a similar change of difference between conformations (of about ~0.4 Å) was observed due to the inclusion of water molecules during molecular docking. Briefly, Figure 11 and Figure 12 show the MIF analysis of non-solvated and solvated conformations obtained with open (Figure 11) and closed (Figure 12) state. Three highly active ligands MK-499, E4031, trimethoprim were used to show important features and their respective distances. However, the least active compound; Pheotribdile is showing features and their respective distances that contribute negatively towards hERG potency (Figure 11 and Figure 12). Table 3 enlists all the features and their positive and negative impact on hERG inhibition and the respective distances between MIF hotspots. The respective differences between distances of MIF created in non-solvated and solvated-open and -closed states are also concluded in Table 3. 

### 2.3. External Test Set Validation

Most recent hERG inhibition experimental data of 30 inhibitors reported in the year 2017 [38,39,40,41] and 2018 [29,42] including malarial compounds tested against hERG from Open Source Malaria (OSM) database [43] was gathered to evaluate the performance of the models (see Appendix A). Compounds with known hERG IC_50_ values were selected for further evaluation and predictions of all four models. Appendix A enlists the SMILE codes of the test data along with experimentally known hERG inhibition potential (pIC_50_) and predicted pIC_50_ values with all four models. R^2^ of each model was calculated to identify the performance and predictive ability. The external validation of all four models showed that the model built using docked conformations of non-solvated-open state exhibits comparatively better R^2^ value of 0.51. Additionally, the results of internal set validation (Table 2) also proved that non-solvated open conformational state model has better predictive ability (R^2^ = 0.58) and performance. Overall the binding energy values of the non-solvated conformations are also better than solvated complexes. The GRIND distances showed that the binding poses within the hERG binding cavity deviate about ~0.4 Å from non-solvated to solvated model conformations. Thus, GRIND analysis also does not favor using solvated conformation for model building. These results are well in line with our molecular docking and PLIF results where solvated complexes contained higher binding energy values and explicit water molecule exert a slight change on conformations that ultimately affect the performance of GRIND model. Overall, our results suggest that the inclusion of water molecule in hERG-binding cavity during molecular docking does not improve the predictive ability of the QSAR model.

## 3. Discussion

In this study, we have investigated the role of water molecules in drug binding of the hERG channel. Our results suggest that the presence of water molecules in the central cavity of hERG channel is involved in water-mediated ligand–protein interactions. However, the presence of water molecules in the hERG-binding cavity does not cause large conformational changes in important functional groups of ligands. We have applied various docking algorithms (including induced fit and genetic algorithms) and scoring function (London dG, GBVI/WSA, and GoldScore) in order to remove any bias. The best model was achieved with conformations obtained with non-solvated-open conformational state cavity. Additionally, we investigate that GBVI/WSA dG function is a comparatively better method to deal where solvated structures are used during molecular docking. Overall, a change of ~0.4Å in conformations of important functional groups of docked poses in solvated and non-solvated models was observed. This minor variation in distance does not significantly affect the performance and predictive ability of the built 3D QSAR GRIND model. Further, internal as well as external test set predictions of the final model suggest that the non-solvated conformations are slightly better than those of the solvated conformations..

The recently determined cryo_EM structure of hERG by Wang and MacKinnon also contains no structural water molecules in the binding cavity [20]. Additionally, the slightly narrow central cavity of hERG compared to other K^+^ channels has greater negative electrostatic potential and this region is less exposed to water [44]. The cryo_EM structure of hERG explained the more negative electrostatic potential of hERG central cavity ~−625 mV comparative to other K^+^ channels ~−125 that might be due to the small volume of the central cavity contributing towards drug trapping. The small volume of high dielectric medium (i.e., water) surrounded by less dielectric medium (i.e., protein) exhibits more electronegative potential that increases the affinity for drug binding [20,44]. Previously reported MD studies showed that water is mainly present outside the membrane, with lower amounts present within the permeation pore to facilitate ion transportation through the pore [45]. Our findings are in line with previous reports that water is not abundantly present in the pore domain of open channel [44,46,47].

It was reported previously that the desolvation effect has a very crucial role during protein–ligand binding in order to determine the structure and free energy of the complex. More specifically, water molecules modulate the binding process by (i) interacting with non-polar groups and contributing to hydrophobic interactions (ii) involved in electrostatic interaction between charged atoms. Moreover, computing solvation energy is a challenge to structure-based drug design because the estimation of binding free energy is a detailed and delicate determination of interactions between (L–R, R–W, L-W, and W–W) inhomogeneous and complicated environment. The issues in current docking studies including protein flexibility, treatment with water molecules and evaluation of binding free energies are not yet solved satisfactory [26].

Herein, the ligand–protein interaction profiling of cryo_EM-docked complexes (non-solvated and solvated) revealed that highly potent inhibitors are not readily interacting with Phe656. However, various studies suggest that Phe656 is critical for drug binding [5,44,48,49]. However, recently Helliwell et al., and Vaz et al., demonstrated that the Phe656 side chains adopt unexpected orientation projecting away from the central cavity and thus, interaction with Phe656 could not be observed in docking studies using recently resolved cryo_EM structure of hERG [48,50]. Additionally, the expected distance between central pore to Phe656 is much greater in cryo_EM open conformation structure [51]. These studies suggest that a small clockwise rotation of the lower part of S6 helix towards the central cavity is required [48,50]. The current study explores the water-mediated interactions between ligand and protein but they do not affect the conformations to a greater extent consequently the predictive ability of the GRIND model is not considerably affected. Thereof, we conclude that the addition of water molecules during the docking process has no significant impact towards the binding of the hERG.

## 4. Materials and Methods 

The aim of this study was to investigate the impact of water in hERG binding cavity towards drug trapping phenomenon. The study was focused to identify the effect of different conformations on the GRIND model. These conformations were obtained after docking of the training set into non-solvated and solvated open and closed conformation at states of hERG binding cavity. The overall workflow is shown in Figure 13. 

The training dataset was used to build the GRIND models; however, the validation of the models was done with the internal and external test set as defined in the Section 2. 

### 4.1. Molecular Docking Simulations

#### 4.1.1. Structure Preparation

In order to perform molecular docking simulation hERG cryo_EM structure (PDB ID: 5Va1) [20] in open conformational state and homology model based on KcsA template [15] in the closed conformational state were used. The structures were prepared and protonated using structure preparation application implemented in MOE (I 2018) followed by energy minimization using the AMBER99 force field [52].

#### 4.1.2. Molecular Dynamic Simulations

In order to explore the conformational stability of hERG cryo_EM complex, Not Another Molecular Dynamic (NAMD) [31] program was used to perform the MD of the cryo_EM complex. The hERG cryo_EM complex was solvated using an explicit TIP3P water box of 1.5 Å boundary dimensions and the temperature was set to 310 k. The periodic boundary conditions were applied to equilibrate the system. The CHARM22 force field [30] was applied for minimization and equilibrations. The system was heated from 0 to 310 k and 1 atm pressure was applied under constant isothermal conditions. The production run of 1 ns was performed. In order to investigate the overall structure stability, the root means square deviation (RMSD) of the Cα atom from its initial coordinates that serve as a reference measure was calculated. The results were analyzed using visual molecular dynamic (VMD) [53].

#### 4.1.3. Solvation of Binding Cavity

After stabilizing the structure the water box was generated around the mutagenesis residues reported in literature these include important drug binding residues from the selectivity filter (Thr623, Ser624, Val625), the S6 helix and inner pore (Gly648, Ser649, Thr652, Phe656) in both conformational states [54] as shown in Figure 1 and Figure 2. The centroid point of the binding cavity exhibiting the coordinate values of X: 75.019, Y: 74.702, and Z: 90.738. The solvated cavity structures were minimized using AMBER99 [52]. The selected area of the binding cavity was solvated in both open and closed state (Figure 1C and Figure 2C).

The molecular docking simulations were performed using MOE (I 2018) and Gold (v.5.3) software in order to generate molecular conformations using different placement methods and scoring functions. The London dG and GBVI/WSA dG scoring function along with Alpha-PMI placement method using induced-fit docking protocol [55] was used to generate four different sets of molecular conformations using MOE software (I 2018). However, GoldScore fitness function in Gold (v.5.3) software was also used to generate four independent sets of docked conformations in non-solvated-open, solvated-open, non-solvated-closed, and solvated-closed binding cavities.

The 100 runs of docking were performed for each non-solvated-open, solvated-open, non-solvated-closed, and solvated-closed binding cavities using MOE (I 2018) and Gold (v. 5.3) software. The GBVI/WSA dG scoring function [56] was used in MOE. It is force field-based scoring function that estimates free energy binding. It was trained using the MMFF94x and AMBER99 force field. The fictional form is a sum of the term as explained in Equation (1):

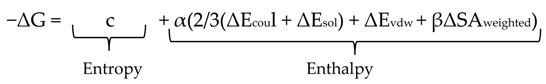
(1)
where:c: represents the average gain/loss of rotational and translational entropy.α, β: are constants which were determined during training (along with c) and are forcefield-dependent. If not using an AMBER forcefield, the parameters will be set by default to the MMFF trained parameters.EcoulEcoul: is the coulombic electrostatic term which is calculated using currently loaded charges, using a constant dielectric of 1.EsolEsol: is the solvation electrostatic term which is calculated using the GB/VI solvation model. For more information on the GB/VI solvation model.Evdw: is the van der Waals contribution to the binding.SAweighted: is the surface area, weighted by exposure. This weighting scheme penalizes exposed surface area.

However, the GoldScore fitness function is made up of four components:External H-bond: protein–ligand hydrogen bond energy,External vdw: protein–ligand van der Waals (vdw) energy,Internal vdw: ligand internal vdw energy,Internal torsion: ligand torsional strain energy.

#### 4.1.4. Final Pose Selection and Ligand–Protein Interaction Analysis

The final pose of each ligand was selected on the basis of least energy score for further ligand-protein interaction analysis. A comparative ligand–protein interaction profile analysis has been performed on selected ligands by lipophilic and ligand efficiency profiling as proposed by Munawar et al. [29]. 

#### 4.1.5. Protein–Ligand Interaction Fingerprints

PLIF technique is used to summarize the interactions between ligand and protein using the fingerprint scheme. In this method, the interaction, that is, hydrogen bond, surface contacts, hydrophobic interactions are classified according to the residues of origin (binding cavity) using fingerprint method that is representative of the training set ligand–protein complexes. All 166 final poses of the training set for open and closed solvated and non-solvated hERG structures were submitted to the PLIF application implemented in MOE (2018) program using default configurations. Non-specific surface contacts were excluded. The generated barcode plots for all four sets of docked conformations summarize the ligand–protein interactions with respect to their solvated or non-solvated structure. 

### 4.2. Grid Independent Descriptors Model Development

The GRIND descriptors of all eight individually loaded conformations were generated in order to compute molecular interaction fields (MIFs). Four different probes including DRY, O, N1 and TIP that represent hydrophobic, hydrogen bond acceptor (carbonyl oxygen), hydrogen bond donor (amide nitrogen) and steric hotspots respectively were used to compute molecular interaction field (MIFs). However total energy was computed as a sum of Lennard-Jones potential (E_lj_), electrostatic energy (E_el_), and hydrogen bond energies (E_hb_) by placing each probe by GRID iteratively as shown in Equation (2).
(2)$$E_{xyz}=\sum E_{lj}+\sum E_{el}+\sum E_{hb}$$

Moreover, most relevant MIF regions were discretized using AMANDA algorithm. The AMANDA algorithm selects all atoms represented in the set of nodes. A node prefiltration step is carried out at the first step by applying suitable energy cutoff values [57]. The discretization step was performed using default energy cutoff values (DRY, −0.75 kcal/mol; O, −0.5 kcal/mol; N1, −4.2 kcal/mol; and TIP, −2.6 kcal/mol) of the respective probes. Nodes representing weak or no interactions are removed in this step. A list of remaining “m_i_” MIF nodes is assigned to every atom (i) of the molecule. A maximum of “n_i_” is selected for atom “i” as shown in Equation (3). (3)$$n_{i}=e^{\left[a,\ln\left(m_{i}\right)\right]}$$

The lowest energy node value is selected. The Euclidean distance was computed between chosen nodes and the rest of the members of the list and added to their field energy values to obtain a simple scoring function as shown in Equation (4).
(4)$$S_{i0}=E_{i}s_{ij}=S_{i\left(j-1\right)}+d_{ik}$$
where “i” is every node and “j” is algorithm step. E_i_ is normalized energy for node i, and d_ik_ is the normalized distance between node “i” and node “k”, selected at (j−1) step. Node with best energy score is selected and the procedure is repeated until n nodes are fully extracted. 

Further, the pre-filtered nodes were encoded by consistently large auto and cross-correlation (CLACC) algorithm [36] that produces more consistent variables. The values obtained for analysis were represented as correlogram plots and used for different chemo-metric analysis. In order to identify the most relevant descriptors that explain the biological activity pIC_50_ values of hERG inhibitors, the partial least square analysis (PLS) was performed. The internal validation was performed using leave one out (LOO) cross-validation [35] method as well as by using internal and external test set.

#### Test Set Validation

Two test sets namely internal and external test set were used for model validation purpose. The internal test set contained 20 percent (41 compounds) of the overall dataset generated using diverse subset selection procedure. For the external test set all compounds tested against hERG reported in the years 2017 [38,39,40,41] and 2018 [29,42], and the compounds available at Open Source Malaria (OSM) database [43] test against hERG liability were also included. In the data set various filters were applied to reduce the data size such as drug-likeness of compounds rule of five, molecular weight (between 200 and 700 KD), electrophysiology assay (patch-clamp), absolute activity values (pIC_50_). Finally, 30 compounds were selected (Appendix A) for the prediction in order to validate the predictive ability of all four models. 

## 5. Conclusions

The impact of water in ligand–protein interactions has remained challenging due to the difficulty in measuring the accuracy and efficiency of docking protocols. Herein, we investigate the impact of explicit water molecules on hERG binding in various conformational states during molecular docking. The GBVI/WSA and GoldScore fitness function were used to generate the binding solutions in solvated and non-solvated open and closed conformational states of hERG. Comparative analysis of different docked conformations used to build the 3D-QSAR GRIND model shows GBVI/WSA scoring function deals with solvated structures more efficiently. Our results show that the addition of water molecules during the docking process has no significant impact on the drug binding on hERG. In comparison, non-solvated conformations represent favorable energy scores. Additionally, PLIF analysis revealed strong hydrogen bond interaction found in the non-solvated binding cavity. Results showed only a small conformational deviation ~0.4 Å of important functional groups was observed where the solvated hERG-binding cavity was used. Furthermore, the internal and external test set prediction of each model also confirms that including water molecule has not significantly affected the model predictions. Overall, neglecting water molecules does not have a significant implication on the computational models for the prediction of hERG inhibition potential but it will help in reducing computational cost and time. The study emphasizes that different binding conformations impact the performance and predictive ability of the 3D-QSAR GRIND model.

## Figures and Tables

**Figure 1 ijms-20-03385-f001:**
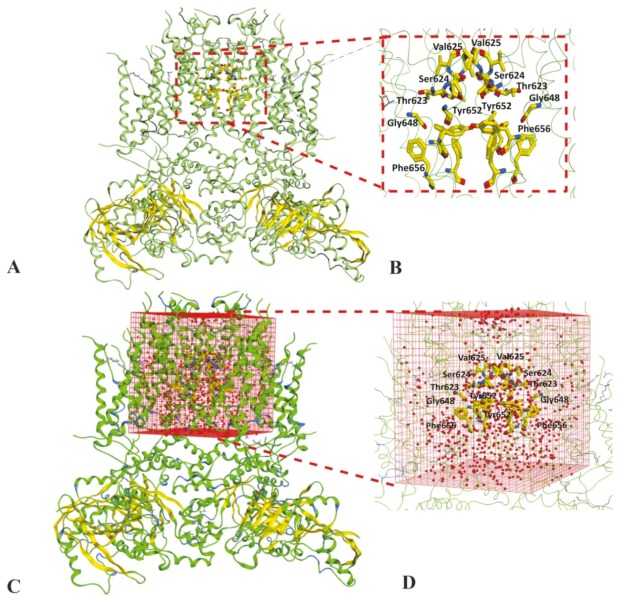
hERG cryo_EM structure in open conformational state (**A**) Binding cavity residues highlighted in non-solvated binding cavity structure. (**B**) Zoom in view of mutagenesis residues of non-solvated-open binding cavity. (**C**) Cryo_EM structure of hERG within simulated water box. (**D**) Zoom in view of solvated-open binding cavity showing mutagenesis amino acid residues and water molecules.

**Figure 2 ijms-20-03385-f002:**
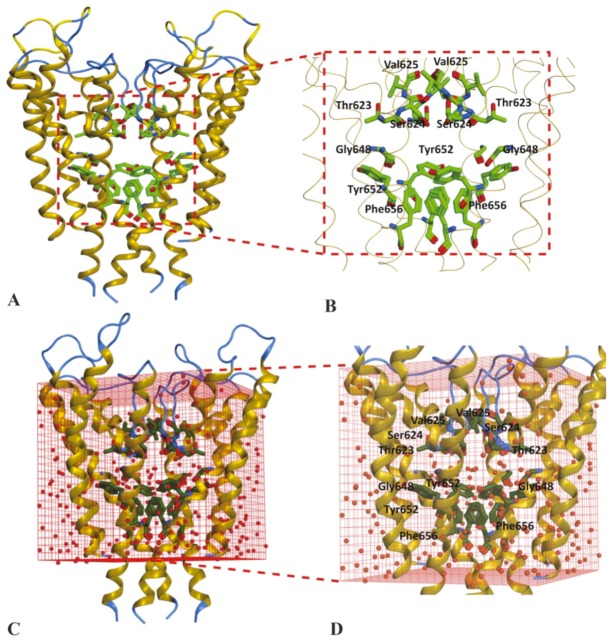
Closed conformational state of hERG homology model based on KcsA template. (**A**) Closed conformational state of homology model with highlighted non-solvated-closed binding cavity. (**B**) Zoom-in view of non-solvated-closed binding cavity showing mutagenesis amino acid residues. (**C**) Solvated-closed conformational state of hERG homology model. (**D**) Zoom-in view of solvated-closed hERG binding cavity showing mutagenesis residues and water molecules.

**Figure 3 ijms-20-03385-f003:**
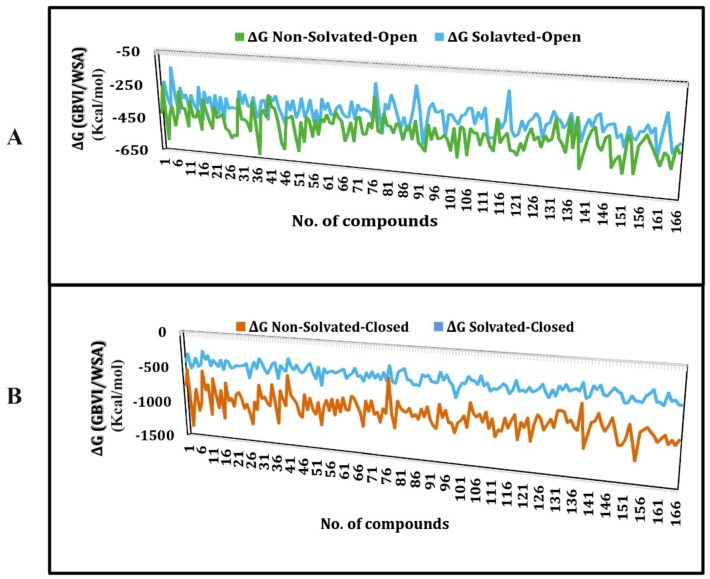
Plot showing binding free energies for the training data. (**A**) Binding free energies of training set docked in non-solvated (green) and solvated cryo_EM structure of hERG in the open conformational state. (**B**) Binding free energies of training set docked in non-solvated (brown) and solvated (blue) closed conformational state homology model of hERG.

**Figure 4 ijms-20-03385-f004:**
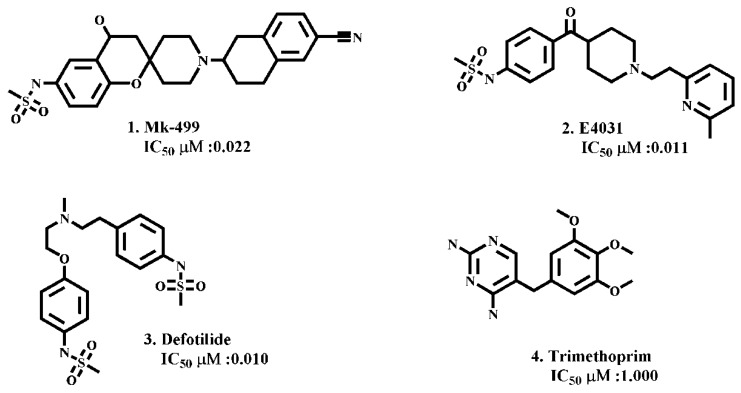
Showing prototype hERG inhibitors (1) Mk-499, (2) E4031, (3) Defotilide, (4) Trimethoprim with best activity lipophilic and ligand efficiency ratios.

**Figure 5 ijms-20-03385-f005:**
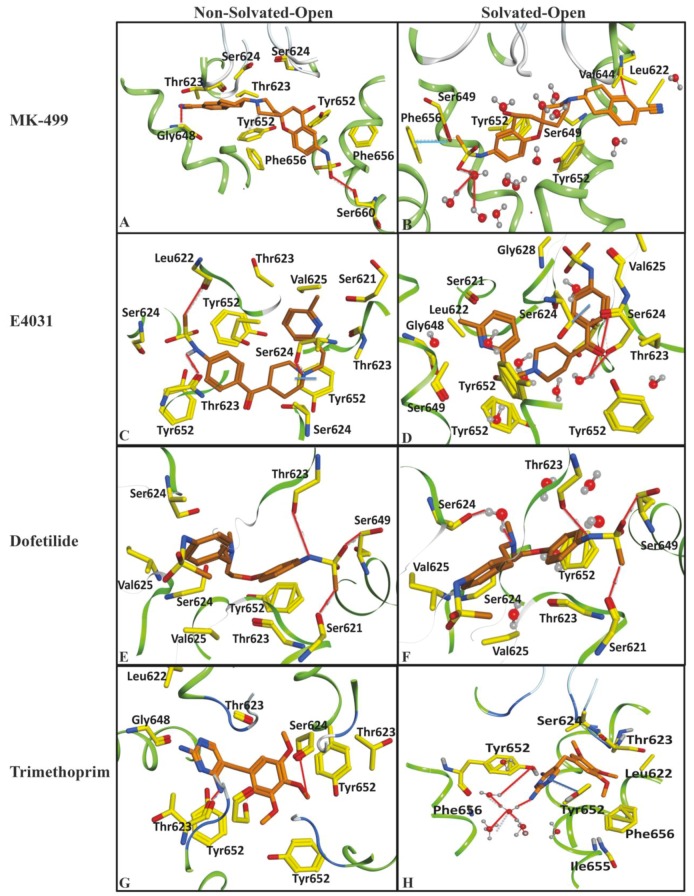
Showing docked ligands in non-solvated open and solvated open binding cavity of hERG cryo_EM structure. (**A**) Final docked pose of Mk-499 in non-solvated-open binding cavity showing hydrogen bond interaction with Gly648, Ser624 and Ser660 and π–π interaction with Tyr652. (**B**) MK-499 showing hydrogen bonding with Leu622 and ligand–water interaction. (**C**) E4031 shows hydrogen bond with Leu622 and Thr623 in non-solvated hERG cavity. (**D**) Docked E4031 final pose showing hydrogen bond and water mediated interactions with hydrophobic interaction with Ser624. (**E**) Dofetilide docked in non-solvated-open binding cavity of hERG showing hydrogen bonding with Ser621 and Thr623. (**F**) Final pose of dofetilide showing water mediated interactions with Ser624 and hydrogen bonding with Ser621 and Thr623. (**G**) Final pose of docked trimethoprim in non-solvated open cavity showing hydrogen bonding with Thr623 and Ser624. (**H**) Trimethoprim final pose docked in solvated open binding cavity showed water mediated hydrogen bond interaction with Tyr652 and π–π interaction with Tyr652.

**Figure 6 ijms-20-03385-f006:**
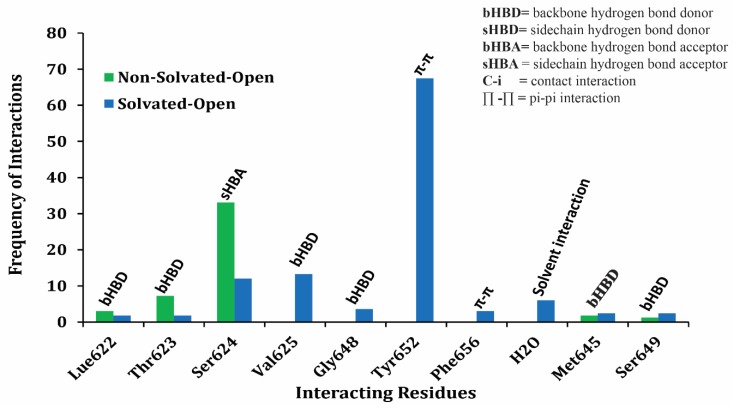
The population histogram summarizing the frequency of occurrence of interactions of ligand–protein interaction profiling between training dataset and non-solvated-open (green) and solvated-open (blue) binding cavity.

**Figure 7 ijms-20-03385-f007:**
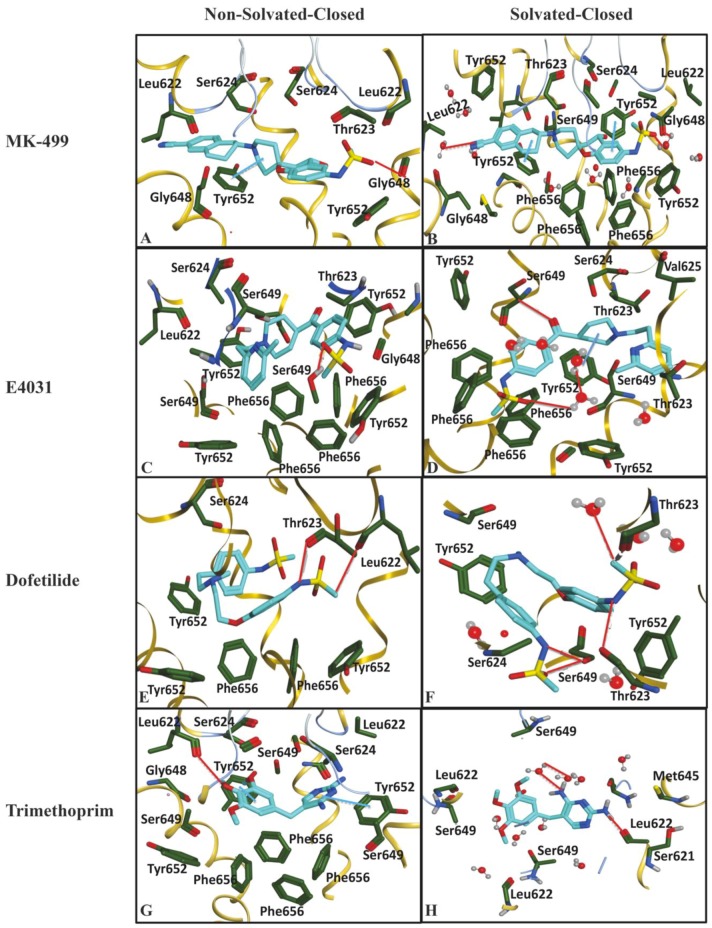
(**A**) Finally selected docked pose of Mk-499 in non-solvated closed binding cavity showing hydrogen bonding with Gly648 and π–π interactions with Tyr652. (**B**) MK-499 showing π–π interactions with Tyr652 and water mediated interactions with Gly648. (**C**) Selected docked conformation of E4031 in non-solvated cavity showed hydrogen bonding with Ser649. (**D**) E4031 in solvated cavity showed hydrogen bonding with Ser649 and oxygen of methanesulfonamide showed ligand–water and water–water hydrogen bond interactions. (**E**) Dofetilide showing hydrogen bonding with Thr623 and Leu622 in non-solvated-closed cavity. (**F**) Dofetilide final docked pose in solvated-closed binding cavity of hERG showing hydrogen bonding with Thr623 and Ser649 the ligand–water hydrogen bond is also observed. (**G**) Final docked pose of Trimethoprim in non-solvated cavity showing hydrogen bonding with Leu622 and π–π interactions with Tyr652. (**H**) Trimethoprim final pose docked in solvated binding cavity showed ligand–water and water–water hydrogen bond interaction with the nitrogen of the Mk-499.

**Figure 8 ijms-20-03385-f008:**
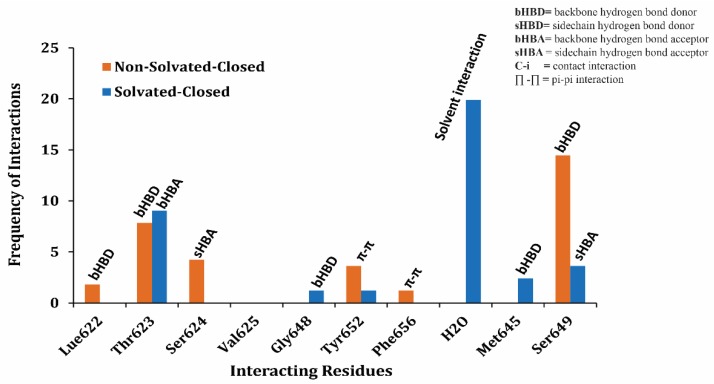
The population histogram summarizes the frequency of occurrence of interactions of ligand–protein interaction profiling between training dataset and non-solvated-closed (orange) and solvated-closed (blue) binding cavity.

**Figure 9 ijms-20-03385-f009:**
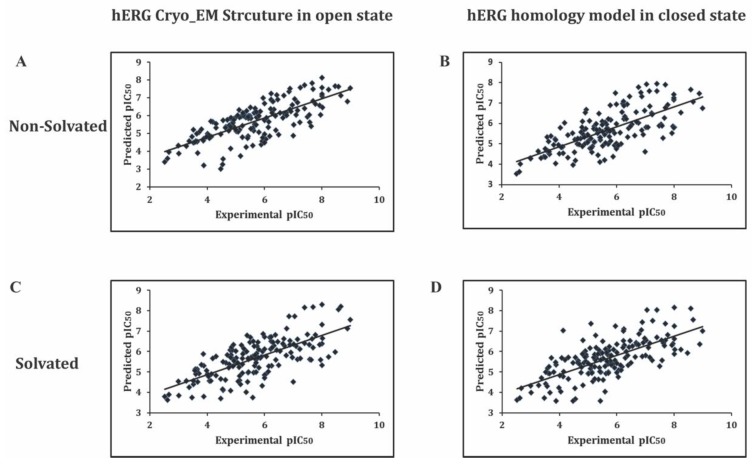
The plot of training set of actual versus observed prediction of (**A**) non-solvated-open state docked conformations and (**B**) non-solvated-closed state docked conformations. (**C**) Solvated-open state docked conformations and (**D**) solvated-closed state docked conformations.

**Figure 10 ijms-20-03385-f010:**
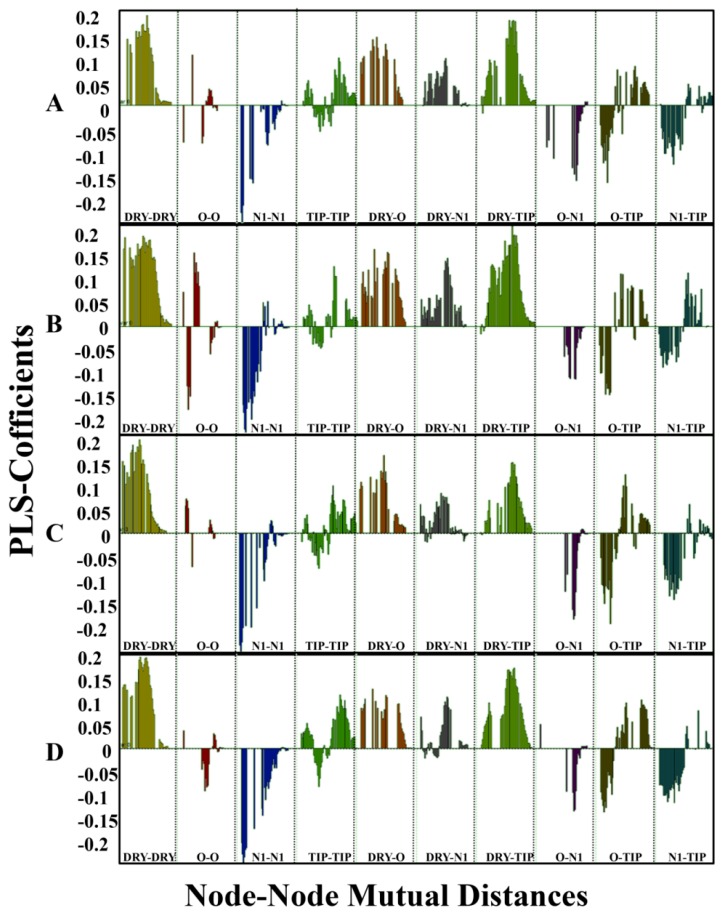
Partial least square (PLS) coefficient plots. (**A**) Non-solvated-open state conformations. (**B**) Solvated-open state conformations. (**C**) Non-solvated-closed conformations. (**D**) Solvated-closed conformations. Indicating DRY-DRY yellow contour, DRY_TIP green contour and DRY-O brown contour present at particular distance increase the tendency of a compound to trap in hERG binding cavity, However, N1-N1 blue contour represents two hydrogen bond donor feature and O-TIP turquoise contour representing one hydrogen bond acceptor and a steric feature present at particular distance impact negatively towards hERG inhibition potential of compounds.

**Figure 11 ijms-20-03385-f011:**
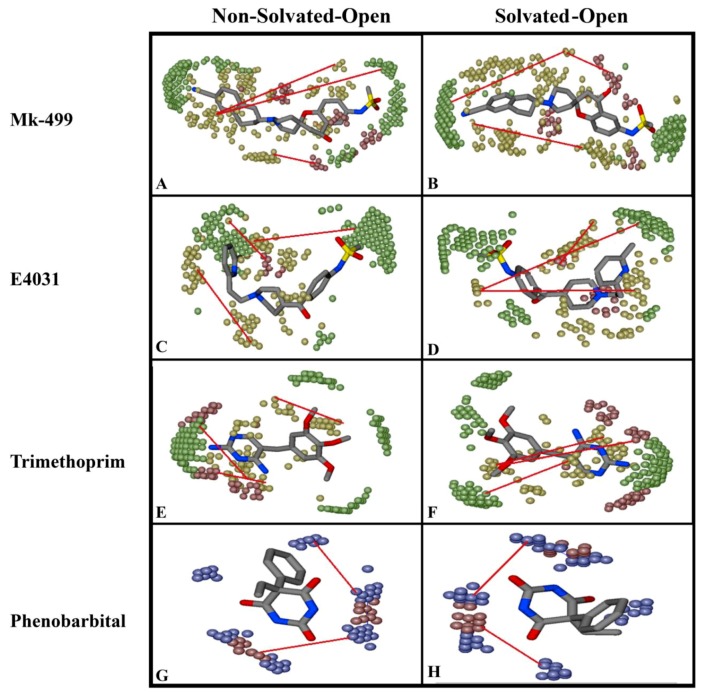
Comparison of Important hotspots regions of non-solvated-open and solvated-open model. Red lines represent the respective distance between the two features that enlist in Table 2. DRY-DRY (yellow hotspots) shows the distance between two hydrophobic features. DRY-TIP yellow and green hotspots respectively shown the distance between hydrophobic feature and steric molecular boundary, DRY-O yellow and red hotspots shows the distance between the hydrophobic group and hydrogen bond donor group that positively contribute the hERG inhibitory potency and the features obtained from solvated and non-solvated conformations are almost similar. (**A**) MK-499 MIF of non-solvated-open state and their respective distances. (**B**) MK-499 MIF of solvated-open state and their respective distances. (**C**) E4031 MIF of non-solvated-open state and their respective distances. (**D**) E4031 MIF of solvated open state and their respective distances. (**E**) Trimethoprim MIF of non-solvated open state and their respective distances. (**F**) Trimethoprim MIF of solvated open state and their respective distances. Likewise (**G**) phenobarbital in non-solvated-open state and their respective distances. (**H**) Phenobarbital in solvated-open state and their respective distances show N1-N1 blue hotspots showing two hydrogen bond acceptor feature and O-NI red and blue hotspot respectively represent present at a particular distance impact negatively towards biological potency of hERG inhibitors.

**Figure 12 ijms-20-03385-f012:**
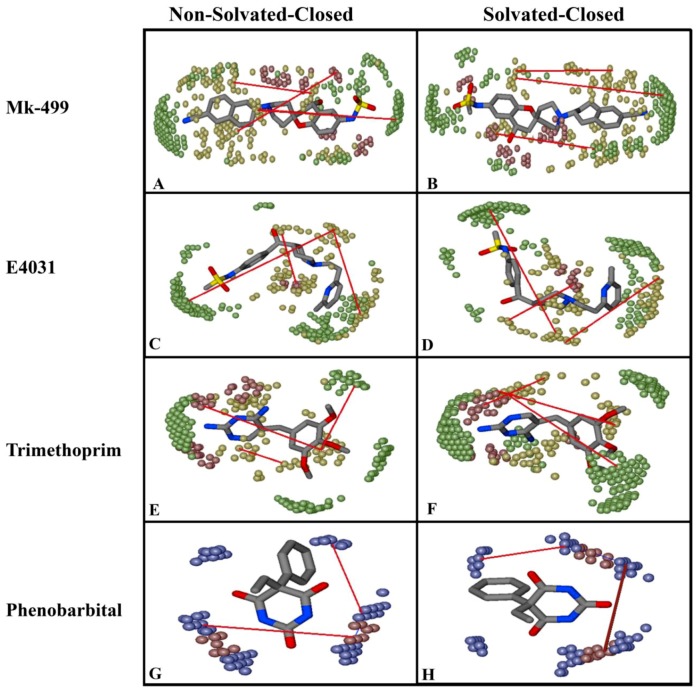
Comparison of important hotspots regions in closed non-solvated and solvated GRIND model. Red lines represent the respective distance between the two features. DRY-DRY (yellow hotspots) shows the distance between two hydrophobic features DRY-TIP yellow and green respectively shown distance between hydrophobic feature and steric molecular boundary, DRY-O yellow and red shows the distance between hydrophobic group and hydrogen bond donor group positively contribute the hERG inhibitory potency and the features obtain from non-solvated and solvated conformations are almost similar. (**A**) MK-499 MIF of non-solvated-closed state and their respective distances. (**B**) MK-499 MIF of solvated-closed state and their respective distances. (**C**) E4031 MIF of non-solvated-closed state and their respective distances. (**D**) E4031 MIF of solvated-closed state and their respective distances. (**E**) Trimethoprim MIF of non-solvated-closed state and their respective distances. (**F**) Trimethoprim MIF of solvated-closed state and their respective distances. Likewise (**G**) phenobarbital in non-solvated-closed state and their respective distances. (**H**) Phenobarbital in solvated-closed state and their respective distances shows N1-N1 blue hotspots shows two hydrogen bond acceptor feature and O-NI red and blue hotspot represent if this feature present at a particular distance will impact negatively towards biological potency of hERG inhibitors.

**Figure 13 ijms-20-03385-f013:**
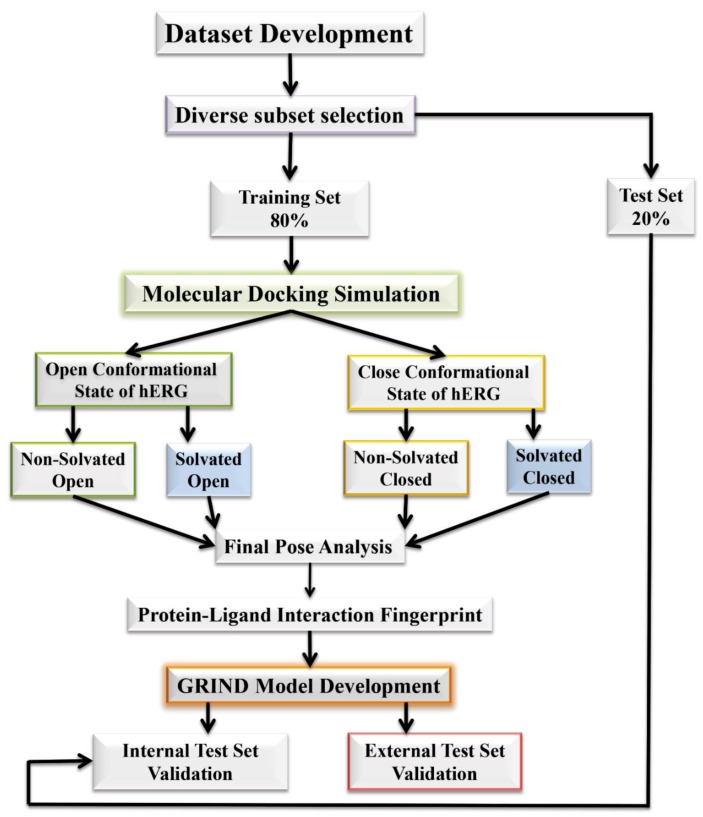
The overall workflow to study the role of explicit water molecules in drug binding of hERG channel and impact of conformational change on the predictive ability of GRIND model.

**Table 1 ijms-20-03385-t001:** Showing energy values (−ΔG) of selected inhibitors in non-solvated and solvated docked complexes in open and closed cavity of hERG.

Compound	−ΔG of Docked Complexes (kcal/mol)		−ΔG of Docked Complexes (kcal/ mol)	
	Non-Solvated-Open	Solvated-Open	Non-Solvated-Closed	Solvated-Closed
MK-499	−388	−372	−949	−467
E4031	−487	−389	−946	−447
Dofetilide	−397	−385	−861	−436
Trimethoprim	−363	−296	−736	−338

**Table 2 ijms-20-03385-t002:** Statistics of the PLS models based on various docked conformations up to first cycle of (Fractional Factorial Design) FFD. Bold values showing final model statistics.

hERG Channel Docked Conformatio-ns	Complete Variable GBVI/WSA Score				FFD 1^ST^ Cycle GBVI/WSA Score				Test Set 1 Validation	Test Set 2 Validation	Complete Variable GoldScore			FFD 1^ST^ Cycle GoldScore					Test Set 1 Validation	Test Set 2 Validation
	LV^a^	q^2^	r^2^	SDEP^b^	LV	q^2^	r^2^	SDEP	R^2^	R^2^	LV	q^2^	r^2^	SDEP	LV	q^2^	r^2^	SDEP	R^2^	R^2^
Non-solvated -open	1	0.43	0.55	0.10	2	0.54	0.62	0.98	0.58	0.51	3	0.35	0.47	1.14	2	0.43	0.52	1.07	0.48	0.41
Solvated -open	2	0.42	0.53	1.07	2	0.44	0.55	1.05	0.52	0.41	2	0.34	0.47	1.14	2	0.36	0.49	1.12	0.32	0.31
Non-solvated-closed	3	0.42	0.62	1.07	2	0.49	0.58	1.00	0.48	0.46	3	0.33	0.42	1.10	2	0.38	0.50	1.02	0.38	0.36
Solvated-closed	1	0.41	0.51	1.07	2	0.43	0.53	1.05	0.36	0.38	2	0.31	0.38	1.12	2	0.36	0.48	1.07	0.31	0.28

^a^ LV= Latent Variables; ^b^ SDEP= Standard Deviation of Error of Prediction.

**Table 3 ijms-20-03385-t003:** Summarizing GRIND features and their mutual distances in hERG open and closed solvated and non-solvated states and their impact on hERG inhibitory potency.

Important Feature	Hotspots Indicating	Impact	Open (cryo_EM) Conformations Distance Å		Closed (Homology) Conformations Distance Å	
			Non-Solvated-Open	Solvated-Open	Non-Solvated-Closed	Solvated-Closed
DRY-DRY	A particular Distance between two hydrophobic moieties	+	12.0–12.4	12.4–12.8	10.0–10.4	10.8–11.2
DRY-TIP	A particular Distance between hydrophobic moiety and steric hot spot	+	13.6–14.0	14.4–14.8	14.8–15.2	15.2–15.6
DRY-O	A particular Distance between hydrophobic moiety and hydrogen bond donor feature	+	7.2–7.6	6.8–7.2	11.6–12.0	12.0–12.4
N1-N1	A particular Distance between two hydrogen bond acceptor groups	-	4.8–5.2	5.2–5.6	4.0–4.4	3.2–3.6
O-N1	A particular Distance between hydrogen bond donor and hydrogen bond acceptor feature	-	16.0–16.4	16.4–16.8	15.6–16.0	16.0–16.4

+ = Positively contributing towards biological activity (pIC_50_); - = Negatively contributing towards biological activity (pIC_50_).

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
