# Peer review of "Molecular Docking Guided Grid-Independent Descriptor Analysis to Probe the Impact of Water Molecules on Conformational Changes of hERG Inhibitors in Drug Trapping Phenomenon"

_ijms, 2019, doi:10.3390/ijms20143385_

Round 1
Reviewer 1 Report
The manuscript describes a computational analysis to probe the impact of water molecules on conformational changes of hERG inhibitors. A lot of work has been done with technical diligence but some topics need further elaboration:
The authors used a cryo-EM structure with 3.7 A of resolution. Probably beyond a simple minimization, a molecular dynamics (MD) simulation would led to a more reliable structure.
Before concluding that the addition of water molecules during the docking process had no significant impact towards the binding of the hERG, different docking software/procedures should be applied and analysed.
The application of MD simulation would allow the authors to study which are the most important water molecules in the binding site, thus focusing their studies on these molecules.
It not clear the selection of GRIND as 3D-QSAR software. This program is independent of the alignment of the compounds; however, as the authors docked the different molecules the final poses could be used as an alignment tool for the 3D-QSAR, further testing the reliability of the docking calculations.
Author Response
We are very grateful to the reviewer for constructive comments to improve the quality of the manuscript.
Following are the point by point reply to the comments
Q1) The authors used a cryo-EM structure with 3.7 A of resolution. Probably beyond a simple minimization, a molecular dynamics (MD) simulation would led to a more reliable structure
MD simulation of the cryo_EM structure was performed which reflect the stability of the structure used for molecular docking. Please see methodology section line 533 to 549 and results in line111-117, also please refer to supporting information Figure S1 and Figure S2
Q2) Before concluding that the addition of water molecules during the docking process had no significant impact towards the binding of the hERG, different docking software/procedures should be applied and analyzed
Different scoring functions including (London DG and GBVI/WSA) along with Alpha-PMI placement method were used to generate different set of molecular conformations using MOE software.
As per suggestion we also generated molecular conformation using gold score fitness function in Gold 5.3 software please see methodology section line 533 to 541. Table 2 line 350 has been added in the revised manuscript that shows the statistics of the GRIND models developed using molecular conformations obtained from Gold and MOE software. Please see methodology line 560-566 GRIND section l line 343-347.
Q3) The application of MD simulation would allow the authors to study which are the most important water molecules in the binding site, thus focusing their studies on these molecules
MD simulation is indeed a good suggestion however, we were using a dataset of 166 compounds and hERG is a complex system, It is a tetramer where a full-length monomer contains 1159 amino acids. Simulation of 166 complexes along with water is computationally too expensive and is not possible currently. Therefore, we use GBVI/WSA model in order to analyze the binding energy and ligand binding conformation and used these conformations for 3D QSAR model development. Statistically good and interpretable 3D QSAR (GRIND) model further strengthen the validity of the binding conformations.
Q4) It not clear the selection of GRIND as 3D-QSAR software. This program is independent of the alignment of the compounds; however, as the authors docked the different molecules the final poses could be used as an alignment tool for the 3D-QSAR, further testing the reliability of the docking calculations
GRIND is alignment independent approach which makes it more efficient for building a 3D QSAR model using structurally diverse data. However, GRIND depends on the 3D conformations of the compounds. Therefore, we used the 3D docked conformations of data set in the open and closed solvated and non-solvated binding cavity were used to build the model. Moreover, docking conformations of structurally diverse data cannot be considered as alignment tool for the 3DQSAR. Thus, the purpose of using docking conformations in GRIND model building was due to GRIND dependency on 3D molecular conformations not alignment. We have elaborated the point in the main text of the manuscript please refer to section (4.4) line 329-335 and line 343-347.
Reviewer 2 Report
The manuscript "Molecular docking guided GRId-INdependent descriptor (GRIND) analysis to probe the impact of water molecules on conformational change of hERG inhibitors in drug trapping phenomenon" is a overall well put together examination of the impact of including explicit water molecules when performing docking with the M.O.E software package. The presentation is fairly straight forward and the results tend to support the author's conclusions. That being said, I do have a few questions that should be address before I can recommend publication:
1) Although the text is largely clear, for non-experts it would be helpful if the author's took the time to expand upon the discussion of GRIND (section 4.4) in more detail (and provide citations). Also please provide a description of how the AMANDA algorithm works.
2) The author's do not describe clearly how the model was solvated. How did they decide where to place the water molecules? Random placement? Random placement and relaxation with fixed protein or non-fixed protein? RISM-Guided? This is my primary concern with the manuscript. The main conclusions about not needing explicit water may be biased by the placement method, i.e. if they solvated in such a way that water is not where it would be in reality, then the waters' impact may not be properly accounted for.
3) Page 7, when comparing energy scores of -487 to 387 kcal/mol the author's say that the score is 100 times higher. This is incorrect, it is simply 100kcal difference, 100 times higher would be if the scores differed by a factor of x100. Please give the paper a once over for grammar and mis-statements (I didn't find many, but this one stuck out)
4) Discussion, paragraph 2, last sentence "Our findings are in line..." This needs a citation.
5) Mutagenesis residues are mentioned on page 5 (just above section 4.2 in the text), please list these for non-experts. Perhaps a figure of the protein with these highlighted would be helpful.
6) Only one scoring fuction GBVI/WSA dG was used. Could you comment on the accuracy of this scoring function and why others in M.O.E where not used? Would your results remain consistent if, for instance, the london-dG scoring was used instead? Also it is somewhat unclear, were the induced-docking options in M.O.E used or just the rigid docking?
Again, I think the work is well presented but with the concerns above I can not currently recommend publication.
Author Response
We are very grateful to the reviewer for constructive comments to improve the quality of the manuscript.
Following are the point by point reply to the comments
Q1) Although the text is largely clear, for non-experts it would be helpful if the author's took the time to expand upon the discussion of GRIND (section 4.4) in more detail (and provide citations). Also please provide a description of how the AMANDA algorithm works.
We have elaborated the point in the main text of the manuscript please refer to section (4.4) line 329-335. GRIND citation is also provided, AMANDA algorithm working is also included please see line 615-629 for more detail reference has also been provided.
Q2) The author's do not describe clearly how the model was solvated. How did they decide where to place the water molecules? Random placement? Random placement and relaxation with fixed protein or non-fixed protein? RISM-Guided? This is my primary concern with the manuscript. The main conclusions about not needing explicit water may be biased by the placement method, i.e. if they solvated in such a way that water is not where it would be in reality, then the waters' impact may not be properly accounted for.
Before docking an MD simulation of the cryo_EM structure was performed in order to get more robust structure for MD simulation. During MD simulation water box was added and allowed to reach at the solvent accessible surface areas within the binding pocket before equilibrium phase. This reflects a random placement within a flexible system. The required information is updated in the revised version of the manuscript; please refer to Line 551-558.
Q3) Page 7, when comparing energy scores of -487 to 387 kcal/mol the author's say that the score is 100 times higher. This is incorrect, it is simply 100kcal difference, 100 times higher would be if the scores differed by a factor of x100. Please give the paper a once over for grammar and mis-statements (I didn't find many, but this one stuck out)
This mistake has been corrected. Please see page 7 line 194 and paper is overall revised for the grammatical and misstatements.
Q4) Discussion, paragraph 2, last sentence "Our findings are in line..." This needs a citation
Required citations have been added. Please see line 494 page 20.
5) Mutagenesis residues are mentioned on page 5 (just above section 4.2 in the text), please list these for non-experts. Perhaps a figure of the protein with these highlighted would be helpful.
Figure 1 and Figure 2 shows the mutagenesis residue in open and closed conformational state of hERG. Suggested information has also been added in section 4.2. Please see line 552-555.
Q6) Only one scoring fuction GBVI/WSA dG was used. Could you comment on the accuracy of this scoring function and why others in M.O.E where not used? Would your results remain consistent if, for instance, the london-dG scoring was used instead? Also it is somewhat unclear, were the induced-docking options in M.O.E used or just the rigid docking?
Different scoring functions including (London DG and GBVI/WSA) along with Alpha-PMI placement method were used to generate different set of molecular conformations using MOE software. However, The GBVI/WSA dG scoring function is discussed in the paper in more detail because it estimates the free energy of binding of the ligand considering the solvation effect which is missing in London DG scoring function. Additionally, molecular conformation generated using gold score fitness function in Gold 5.3 software are also included in the revised manuscript. Please see the Methodology Line 560-590. Induced fit docking protocol implemented in MOE was used for molecular docking simulation while the genetic algorithm was used in GOLD.
Reviewer 3 Report
This article addresses the importance of internal waters in ligand binding to the potassium channel hERG. Based on their docked ligand conformations, the authors present a predictive model for estimating the inhibitory efficiency of small molecule inhibitors. The authors have predicted the docked conformations of a number of inhibitors and calculated their binding free energy. They have characterized the protein-ligand interactions using a set of molecular features, based on which, they have developed their predictive models. This work may help in the design of future inhibitors. Here are the major comments:
A major part of the manuscript deals with the impact of cavity waters on the binding of small molecules. However, their methodology of modeling the water in the hERG binding cavity raises question. While it is not clear how exactly the water molecules were placed inside the cavity, it appears to be a random placement, followed by minimization. Given the dynamic nature of water, in my opinion, this approach is insufficient to capture the energetics of local water. The water box needs to be equilibrated and the position of the water molecules should be sampled using MD simulations with fully flexible protein and ligand. Then multiple snapshots from the MD simulations should be used to calculate the average binding free energy using the GB model. Such a thorough approach may not be suitable for the several hundred ligands the authors have studied, but can be followed for a few selected ligands. Right now, the binding energy comparison with and without water using the single minimized structure per ligand is not reliable in my opinion. My suggestion would be to omit the discussion of water, unless the authors can follow a suitable approach to address it. The authors can consult the following article to get a better understanding of the nuances of binding free energy calculation using implicit solvation:
Samuel Genheden & Ulf Ryde (2015) The MM/PBSA and MM/GBSA methods to estimate ligand-binding affinities, Expert Opinion on Drug Discovery, 10:5, 449-461
The explanation of entropy as the reason for the worse binding free energy in presence of water sounds a bit hand waving, since only the vibrational entropy of the cavity water is calculated. The rotational and translational entropy are not taken into account. On the other hand, the worse binding energy could also be due to the lack of sufficient equilibration of the water molecules, which sounds more plausible to me.
Since this is an ion channel, the electrostatics of the K+ ions in the channel are likely to influence ligand interaction with the protein. How was the influence of such ions taken into account?
Due to the lack of validation against known crystal structures of inhibitor bound hERG, the docked structures discussed here could be (at least some of them) inaccurate. While this does not necessarily disqualify their work, I think it is important to mention this in the manuscript.
Compared to experimentally observed binding free energies of small molecules, the binding energy scale in the manuscript seems too unphysical. Some of the energy values are lower than -1000 kcal/mol. While this is likely due to the use of the GB model, which approximates the solvation free energy and hence only qualitatively reproduces the energy values, rather than quantitatively, it is important to discuss this in the manuscript.
Author Response
We are very grateful to the reviewer for constructive comments to improve the quality of the manuscript.
Following are the point by point answer to the comments
Q1) A major part of the manuscript deals with the impact of cavity waters on the binding of small molecules. However, their methodology of modeling the water in the hERG binding cavity raises question. While it is not clear how exactly the water molecules were placed inside the cavity, it appears to be a random placement, followed by minimization. Given the dynamic nature of water, in my opinion, this approach is insufficient to capture the energetics of local water. The water box needs to be equilibrated and the position of the water molecules should be sampled using MD simulations with fully flexible protein and ligand. Then multiple snapshots from the MD simulations should be used to calculate the average binding free energy using the GB model.
Before docking an MD simulation of the cryo_EM structure was performed in order to get more robust structure for MD simulation. During MD simulation water box was added and allow reaching at the solvent accessible surface areas within the binding pocket before equilibrium phase. This reflects a random placement within a flexible system. The required information is updated in the revised version of the manuscript, please refer to Line 540-549.
Q2)Such a thorough approach may not be suitable for the several hundred ligands the authors have studied, but can be followed for a few selected ligands. Right now, the binding energy comparison with and without water using the single minimized structure per ligand is not reliable in my opinion. My suggestion would be to omit the discussion of water, unless the authors can follow a suitable approach to address it. The authors can consult the following article to get a better understanding of the nuances of binding free energy calculation using implicit solvation
Before docking an MD simulation of the cryo_EM structure was performed in order to get more robust structure for MD simulation. During MD simulation water box was added and allowed to reach at the solvent accessible surface areas within the binding pocket before equilibrium phase. This reflects a random placement within a flexible system please refer to Line 551-58.
However, simulation of a dataset of 166 compounds and hERG (1159 amino acids along with water is computationally too expensive and is not possible currently. The simulation of selected ligand-protein complexes could be a good option however, in this approach most probable binding conformations of all 166 complexes were required rather than selected ligands as the GRIND was developed using binding conformations of whole data set. Therefore, we use GBVI/WSA model in order to analyze the binding energy and ligand binding conformation and used these conformations for 3D QSAR (GRIND) model development. Statistically good and interpretable 3D QSAR (GRIND) model further strengthen the validity of the binding conformations.
The ultimate goal of this manuscript is to check the performance (predictive ability) of 3D QSAR model built using different set molecular conformations in solvated and non-solvated scenarios of hERG cavity. This is only for hERG due to its unique structural and functional properties, complexity and presence of water filled basal cavity at the entrance of the protein and thus, cannot be generalized for other systems.
Q3) Samuel Genheden & Ulf Ryde (2015) The MM/PBSA and MM/GBSA methods to estimate ligand-binding affinities, Expert Opinion on Drug Discovery, 10:5, 449-461
The MM/PBSA and MM/GBSA are good methods for explicit solvent simulations and its post-processing however reports showed that GBVI model have similar (Hosen, Rubayed et al. 2018) or better results than MM-GBSA(Aldulaijan and Platts 2012). The implementation of MM-GBSA and combining with MM/PBSA is comparatively expensive protocol and time demanding for this study. However, comparison of binding free energies of prototype inhibitors of hERG using different scoring functions including solvation effect and MD simulation of resultant complexes will be submitted in a follow-up study.
Please see:
Aldulaijan, S. and J. A. Platts (2012). "Prediction of Peptide Binding to Major Histocompatibility II Receptors with Molecular Mechanics and Semi-Empirical Quantum Mechanics Methods." Journal of Molecular Biochemistry 1(1).
Hosen, S. Z., et al. (2018). "Prospecting and Structural Insight into the Binding of Novel Plant-Derived Molecules of Leea indica as Inhibitors of BACE1." Current pharmaceutical design 24(33): 3972-3979.
Q4) The explanation of entropy as the reason for the worse binding free energy in presence of water sounds a bit hand waving, since only the vibrational entropy of the cavity water is calculated. The rotational and translational entropy are not taken into account. On the other hand, the worse binding energy could also be due to the lack of sufficient equilibration of the water molecules, which sounds more plausible to me.
Rotational and translational entropy are also considered please refer to methodology of GBVI line 575. Before docking an MD simulation of the cryo_EM structure was performed in order to get more robust structure for MD simulation. We performed MD simulation of 1ns of cryo_EM structure to attain the structure stability. During MD simulation water box was added and water was allowed to reach the solvent accessible surface areas within the binding pocket before equilibrium phase. This reflects a random placement of water within a flexible system. The required information is updated in the revised version of the manuscript, please refer to Line 551-58.
Q5) Since this is an ion channel, the electrostatics of the K+ ions in the channel are likely to influence ligand interaction with the protein. How was the influence of such ions taken into account?
hERG is potassium selective ion channel that efflux the potassium ions out of cardiac myocyte, any blockade either inherited or drug-induced inhibit the flow of ions. Untill very recent, there is no evidence of interaction between ligands and K+ ion in the literature. The underlying drug trapping mechanism is still not known clearly due to the promiscuous nature of hERG cavity. The only evidence of the inhibition of K+ current cause QT interval prolongation on the surface of the electrocardiogram is explained. Please see the introduction section line 44-50.
Q6) Due to the lack of validation against known crystal structures of inhibitor-bound hERG, the docked structures discussed here could be (at least some of them) inaccurate. While this does not necessarily disqualify their work, I think it is important to mention this in the manuscript.
The suggested point has been included in the main manuscript please see introduction section line 74-75.
Q7) Compared to experimentally observed binding free energies of small molecules, the binding energy scale in the manuscript seems too unphysical. Some of the energy values are lower than -1000 kcal/mol. While this is likely due to the use of the GB model, which approximates the solvation free energy and hence only qualitatively reproduces the energy values, rather than quantitatively, it is important to discuss this in the manuscript.
7) The suggested information has been updated in the main manuscript please refer to page 7 Line 202-207
Reviewer 4 Report
The presented research is focused on one of the oldest target of in silico studies, the hERG otassium channel blockade. Considering the large number of articles dealing with this issue, the originality of the article is not very high, but it does add something new.
The methods performed and described seem to be correct, but there are some formatting and styling mistakes that need a closer attention (see the tables, for example).
Check the abstract. It seems to be longer than the journal’s requirements
Row 44. Some important examples of drugs should be provided. It would be easier for the readers.
Row 67. Describe also other types of in silico methods to flag hERG problems. As far as I know, many software applications have such a function (for example ADMET predictor, Derek). A small discussion should be added on these methods.
Row 158. The manuscript should present the exact criteria for choosing MK-499, E4031, Dofetilide and Trimethoprim as prototypes. Considering MK-499, and all the similar cases of optical isomerism, was MK-499 docked as R,R-isomer? The same question for all compounds. Figure 4 does not show stereobonds and it should be corrected.
The conclusion section should be shorter and clear. The discussion section seems too long. It should not repeat sections described in the results or the conclusion. What is the future impact of these data? How can other authors use this research in the future?
Author Response
We are very grateful to the reviewer for constructive comments to improve the quality of the manuscript.
Following are the point by point answer to the comments
Q1) The methods performed and described seem to be correct, but there are some formatting and styling mistakes that need a closer attention (see the tables, for example).
Suggested changes have been incorporated in the main manuscript.
Q2) Check the abstract. It seems to be longer than the journal’s requirements
Abstract has been updated.
Q3) Row 44. Some important examples of drugs should be provided. It would be easier for the readers
Done please see line 43-44.
Q4) Row 67. Describe also other types of in silico methods to flag hERG problems. As far as I know, many software applications have such a function (for example ADMET predictor, Derek). A small discussion should be added on these methods.
Done see line 64-69.
Q5) Row 158. The manuscript should present the exact criteria for choosing MK-499, E4031, Dofetilide and Trimethoprim as prototypes. Considering MK-499, and all the similar cases of optical isomerism, was MK-499 docked as R,R-isomer? The same question for all compounds. Figure 4 does not show stereobonds and it should be corrected.
Since there is no evidence of stereoselectivity towards hERG in literature, the racemic mixture of MK-499 was used. The prototypes compounds are the model substrates used in binding assays i.e Dofetilide and state dependency studies of hERG(Diaz, Daniell et al. 2004).
Please see:
Diaz, G. J., et al. (2004). "The [3H] dofetilide binding assay is a predictive screening tool for hERG blockade and proarrhythmia: Comparison of intact cell and membrane preparations and effects of altering [K+] o." Journal of pharmacological and toxicological methods 50(3): 187-199.
The conclusion section should be shorter and clear. The discussion section seems too long. It should not repeat sections described in the results or the conclusion. What is the future impact of these data? How can other authors use this research in the future?
6) The conclusion has been revised. The discussion has been updated.
The future prospect has been updated please see line 658-61.
Round 2
Reviewer 1 Report
The paper can be accepted for publication.
Author Response
We would thank the reviewer for previous suggestions and we believe that addressing these points has improved the quality of the manuscript. We have also revised the latest version carefully and all English grammatical mistakes are corrected. We thank the reviewer for accepting this manuscript for publication.
Reviewer 2 Report
The authors have addressed my previous concerns and I have no more issues with the manuscript. A few sentences are still a little "stilted" and could some revision; however, after a quick grammatical review it should be ready for publication.
Author Response
We would thank the reviewer for the previous suggestions and we believe that addressing these points has improved the quality of the manuscript. We have also revised the latest version carefully and all English grammatical mistakes are corrected.
Reviewer 3 Report
In general, the authors responses are to my satisfaction. However, some english correction are still necessary, specially in the new sections added in red font.
Author Response
We would thank the reviewer for the previous suggestions and we believe that addressing these points has significantly improved the quality of the manuscript. We have revised the entire manuscript carefully including the conclusion in the latest version and all English grammatical mistakes are corrected especially in the newly added sections.